# Asymptomatic Malaria Reservoirs in Honduras: A Challenge for Elimination

**DOI:** 10.3390/pathogens13070541

**Published:** 2024-06-27

**Authors:** Sharon Banegas, Denis Escobar, Alejandra Pinto, Marcela Moncada, Gabriela Matamoros, Hugo O. Valdivia, Allan Reyes, Gustavo Fontecha

**Affiliations:** 1Instituto de Investigaciones en Microbiología, Facultad de Ciencias, Universidad Nacional Autónoma de Honduras, Tegucigalpa 11101, Honduras; 2Department of Parasitology, U.S. Naval Medical Research Unit South (NAMRU SOUTH), Lima 07006, Peru; 3Unidad de Entomología, Región Sanitaria de Gracias a Dios, Secretaría de Salud de Honduras, Puerto Lempira 33101, Gracias a Dios, Honduras

**Keywords:** asymptomatic malaria, PET-PCR, nPCR, rapid diagnostic tests, Honduras, elimination

## Abstract

Background: Efforts on a global scale for combating malaria have achieved substantial progress over the past twenty years. Two Central American nations have accomplished their goal of eliminating malaria: El Salvador and Belize. Honduras has decreased the incidence of malaria and now reports fewer than 4000 malaria cases annually, aspiring to reach elimination by 2030. To accomplish this goal, it is essential to assess the existing strategies employed for malaria control and to address the task of incorporating novel intervention strategies to identify asymptomatic reservoirs. Methods: A survey for detecting asymptomatic cases was carried out in the community of Kaukira, in Gracias a Dios, Honduras, focusing on malaria transmission during 2023. Asymptomatic community members were recruited as participants, malaria screening was performed through a rapid diagnostic test in situ, and a blood sample was collected on filter paper. Highly sensitive molecular assays based on photo-induced electron transfer PCR (PET-PCR) were performed to detect the two species of *Plasmodium* circulating in Honduras: *Plasmodium vivax* and *Plasmodium falciparum*. In addition, the identification of the parasite species was verified by amplifying three genetic markers (*Pvmsp3α*, *Pvmsp3ß*, and *Pfmsp1*). Results: A total of 138 participants were recruited, mostly adult women. All individuals tested negative on the rapid diagnostic test. Positive results for malaria were detected by PET-PCR in 17 samples (12.3%). Most samples (12 out of 17) were amplified with a Ct value between 37 and 42, indicating very low parasitemias. Out of the 17 samples, 16 of them also showed amplification in the species assays. There were nine cases of *P. falciparum* infections and seven cases of *P. vivax* infections that were further confirmed by nested PCR (nPCR) of *Pvmsp3* and *Pfmsp1*. Parasitemias ranged from 100 p/μL to less than 0.25 p/μL. One sample showed mixed infection. Conclusions: The existence of asymptomatic malaria reservoirs in Honduras can contribute to disease transmission and pose a challenge that may hinder elimination efforts, requiring public health authorities to modify surveillance strategies to identify the disease and treat this population accordingly.

## 1. Background

Significant advancements have been achieved in controlling malaria over the past two decades. The number of countries that recorded less than 100 malaria cases per year has increased from six in 2000 to twenty-seven in 2022. Also, the number of countries with less than ten indigenous cases increased from four in 2000 to twenty-five in 2022 [1]. Malaria elimination has been accomplished in a growing number of countries. In 2023, the World Health Organization (WHO) certified three countries (Azerbaijan, Belize, and Tajikistan), as malaria-free, and several others are making progress towards eliminating the disease by 2024 [1]. Belize has become the fourth country in the WHO Region of the Americas, and the second in Central America, after El Salvador, to reach this status within the last 5 years.

Still, the situation is less favorable in the remaining countries of the Central American isthmus. Case incidence in Costa Rica, Nicaragua, and Panama experienced a rise of 55% or higher in 2022 when compared to 2015. However, in 2022, Honduras reduced the incidence of malaria infections by 25% to 55% compared to 2015, and the country has officially entered the elimination stage [2].

Following the WHO guidelines, localities with moderate and high transmission of malaria are defined by an annual parasite index (API) of 1 to 10 cases per 1000 population and over 10 cases per 1000 population, respectively. Areas of low transmission are defined as those with an API rate < 1 per 1000 people [3]. Based on this classification, Honduras is categorized as a low transmission country, with an API of 0.25 [4]. However, several locations along the Atlantic coast, such as the department of Gracias a Dios, are classified as high transmission areas, with an API beyond 20 per 1000 residents. In 2022, Gracias a Dios reported 3476 malaria cases (65% by *P. vivax*), accounting for 97% of the total malaria cases in the country [4]. Moreover, specific communities, such as Kaukira in 2023, have even higher incidence rates, with an API of 79.9 [4].

Honduras collaborates with the technical assistance of the Pan American Health Organization (PAHO), as well as other key partners, including the Inter-American Development Bank (IDB), Clinton Health Action Initiative (CHAI), and the Global Fund to eliminate malaria. Current disease control programs rely heavily on readily available detection methods like light microscopy (LM) and rapid diagnostic tests (RDT). These methods primarily identify individuals showing symptoms and seeking medical care, or those uncovered through targeted efforts in known infection hotspots. However, this approach misses two crucial groups: asymptomatic carriers (with parasites in the peripheral blood in the absence of fever or other acute symptoms in individuals who have not received recent anti-malarial treatment) and symptomatic individuals who do not seek medical care (often due to access limitations). These overlooked individuals serve as hidden reservoirs, as they may have gametocytes circulating in the peripheral blood, thus maintaining the transmission of the disease [5].

A recent study was conducted by our research group, at Puerto Lempira, located in the Gracias a Dios department, to evaluate the sensitivity of LM compared to two molecular techniques [6]. The authors reported that molecular techniques, specifically a photo-induced electron transfer PCR (PET-PCR), were 1.6 times more sensitive than LM. They also observed that 14 patients who tested positive by PET-PCR were not detected by LM, even though the participants showed symptoms of malaria. Ultrasensitive techniques, like PET-PCR, have demonstrated their utility as effective instruments for detecting asymptomatic or subclinical malaria, particularly in the context of surveillance surveys [7,8,9,10,11,12].

Considering the country’s entry into the elimination stage, this study aimed to determine the prevalence of asymptomatic malaria infections in the region of La Moskitia of Honduras, through an active search strategy.

## 2. Methods

### 2.1. Ethical Consideration

The study was approved by the Ethics Committee of the Infectious and Zoonotic Diseases Master’s Program (CEI-MEIZ) at the Universidad Nacional Autónoma de Honduras (Protocol 02-2023). Before patients were enrolled in the study sites, benefits and any perceived risks were explained to all participants in Spanish or the Miskito language. Written informed consent was obtained from all subjects involved in the study and/or their legal guardians in the case of minors.

### 2.2. Study Site and Population

This study was conducted as a cross-sectional survey for asymptomatic malaria cases. Asymptomatic infections were described as the presence of parasites (antigens or DNA) in the peripheral blood in the absence of fever or other acute symptoms in individuals who have not received recent anti-malarial treatment [13,14]. The survey was carried out in the department of Gracias a Dios, Honduras, in the village of Kaukira, a municipality of Puerto Lempira (15.313611, −83.583056/15.309837, −83.567727). Kaukira is a remote region, that can be accessed by boat only. It is located a few meters above sea level and the population consists mainly of Miskito people. Living standards are generally based on a subsistence economy including fishing, agriculture, and occasional hunting. Households of two neighborhoods were visited (Kiankan and Cocodacra) (Figure 1). Kiankan includes 128 homes and 590 inhabitants, and Cocodacra includes 217 homes and 1025 inhabitants, according to the local census from 2022. Puerto Lempira is characterized as one of the rainiest regions of the country with up to 2167 millimetres of precipitation on average per year. Precipitation is unevenly distributed throughout the year, except for January and May to December, which have rainfall exceeding 100 mm. The average annual temperature fluctuates between 23 °C and 32 °C, rarely falling below 22 °C or rising above 33 °C. The Kaukira community was selected for the survey due to the high incidence of malaria in 2023, as reported by the national malaria surveillance system (https://www.salud.gob.hn/sshome/index.php/malaria, accessed on 23 March 2024), with a total of 762 cases reported during 2023.

### 2.3. Sample Collection and RDT

The survey occurred at the household level, where individuals of any age and sex participated. Participant recruitment took place between 8 and 17 November 2023. Households were selected randomly across the community and individuals of any age and sex were invited to participate. Inclusion criteria for study participants were the ability to provide informed consent. Three study teams visited households to screen for eligibility, obtained informed consent, administered questionnaires, and performed finger-prick blood collection. The survey registered individual demographic information, history of malaria infection, recent travel information, and history of having lived with individuals diagnosed with malaria in the last 30 days.

An RDT (Bioline™ Malaria Ag P.f/P.v, Abbott, Chicago, IL, USA) was performed using blood collected from a finger by lancet puncture according to manufacturer instructions. Additionally, approximately 50 µL of blood was collected on InstaDNA Card^®^ (Himedia, Kelton, PA, USA) filter paper for further molecular analysis.

A Fisher exact test was conducted to examine whether there was a significant association between individuals who tested positive and the demographic variables.

### 2.4. DNA Extraction and PET-PCR

The blood samples were stored on filter paper and placed in separate sealed plastic bags with desiccant. The samples were then sent to the “*Centro de Investigaciones Genéticas*” of the Universidad Nacional Autónoma de Honduras in Tegucigalpa. Three circles of 1.2 mm^2^ each were cut from paper impregnated with blood for DNA extraction. The DNA was extracted with the Extracta DNA Prep for PCR kit (Quantabio, Beverly, MA, USA) according to manufacturer instructions.

Each sample was amplified three to four times using specific primers for the genus *Plasmodium* by PET-PCR [6,7,15]. Samples that returned at least one positive result for the genus were subjected to amplification using two distinct pairs of primers targeting the two parasite species found in Honduras (*P. falciparum* and *P. vivax*). Amplification reactions were carried out in a volume of 20 μL and contained 10 µL Go Taq^®^ Probe qPCR Master Mix (Promega Corp. Madison, WI, USA), 0.5 μL of each primer (10 μM) (Table 1), 4 μL of nuclease-free water, and 5 μL of DNA (~40 ng/µL). Reactions were run on a Mic qPCR Cycler (Bio Molecular Systems, Brisbane, Australia) and the results were visualized in the Mic qPCR Cycler Software v.2.10.1. The amplification conditions for both genus and species detection were 95 °C for 15 min, 45 cycles at 95 °C for 20 s, 63 °C for 40 s, and 72 °C for 30 s. The fluorescence channel was selected for each labeled primer (6FAM for genus and HEX for species). Each experiment comprised both positive and negative controls. PET-PCR reactions for both parasite species were performed in duplicate. A cycle threshold (Ct) of 42 or below was used to consider samples as positive. Samples with a Ct value between 42 and 44 were also considered putative positive with very low parasitemia. To prevent false positives, the amplification curve’s correct sigmoidal shape was examined in every case (Appendix A).

The number of parasites per μL of blood was quantified using a reference curve. The reference curve was prepared by consecutive serial dilution (from 1:1 to 1:1,000,000) in nuclease-free water of DNA extracted from a quantified *P. falciparum* strain 3D7 containing 100,000 parasites per μL. This lyophilized DNA was donated by the Malaria Branch of the CDC in Atlanta, GA, USA. Using this standard curve, the number of parasites per μL present in the sample was estimated based on the Ct value (Appendix A).

To verify the presence of *P. vivax* in the positive samples, a nested PCR (nPCR) method that targeted the *Pvmsp3α* and *Pvmsp3ß* genes was conducted [20]. *Pvmsp3α* was amplified using a final volume of 25 μL. The mixture consisted of 12.5 μL of KOD DNA Polymerase Master Mix (Sigma-Aldrich, Darmstadt, Germany), 1.0 μL of each primer (10 μM) (Table 1), 9.5 μL of nuclease-free water, and 1 μL of DNA. A subsequent nested reaction was conducted using identical concentrations to the previous one. The amplification conditions for both reactions consisted of an initial denaturation step at 98 °C for 30 s, followed by 35 cycles of denaturation at 98 °C for 10 s, annealing at 54 °C for 10 s, and extension at 68 °C for 30 s. The reactions concluded with a final extension step at 68 °C for 2 min. After the visualization of amplification bands in the nested reaction, the size of the amplicon was registered. *Pvmsp3ß* was amplified using a reaction mixture consisting of 12.5 μL of KOD DNA Polymerase Master Mix (Sigma-Aldrich, Darmstadt, Germany), 1.0 μL of each primer (10 μM) (Table 1), 8 μL of nuclease-free water, and 2.5 μL of DNA. A nested reaction was conducted using identical concentrations as the initial one, except that just one microliter of DNA and 9.5 μL of nuclease-free water were used. The amplification program was the same as for *Pvmsp3α*.

Samples positive for *P. falciparum* were confirmed by amplification of the *Pfmsp1* gene by a nPCR as previously described [18,19]. Primary and secondary PCRs were performed in a total volume of 50 μL containing 25 μL of KOD polymerase and 10 μL and 1 μL of DNA, respectively. Three secondary and independent PCRs were run for each allelic family using specific oligonucleotides (Table 1). Cycle conditions for primary PCR were 98 °C for 30 s; 30 cycles of 98 °C for 10 s, 54 °C for 10 s, and 68 °C for 30 s; and a final extension at 68 °C for 2 min. One μL of primary PCR product was amplified in the two nested PCR reactions with the following cycle conditions: 98 °C for 1 min; 35 cycles of 98 °C for 10 s, 58–59 °C for 10 s, and 68 °C for 30 s; and one final step of 68 °C for 2 min. Separate reactions were required to amplify one out of three possible allelic families: K1, MAD20, and RO33.

## 3. Results

A total of 138 participants were enrolled from 81 households. Most subjects were female (*n* = 91, 65.9%). The mean age was 41.8 years (standard deviation = 18.2; median = 40) and most of them were homemakers (*n* = 77, 55.8%).

Most subjects did not travel outside the community within the last three months (*n* = 119, 86.2%), eight reported a previous malaria case in the last twelve months (5.8%), and twenty-eight reported a previous case in the household during the last month (20.3%).

All the participants tested negative for malaria using RDT. However, 17 individuals (12.3%) tested positive for the presence of the *Plasmodium* genus using PET-PCR amplification, with a Ct value of 44 or below (Appendix A). Twelve (8.7%) of these seventeen samples returned a Ct value of 42 or less. The mean Ct value of all positive samples for the genus *Plasmodium* spp. was 40.6 [34.8–44.1], with a standard deviation of 2.7. Fourteen samples, including those with Ct values over 42, were positive for the markers *Pvmsp3α*, *Pvmsp3ß*, and *Pfmsp1*, thereby confirming the presence of infections with extremely low parasitemias (Figure 2).

The mean age of cases positive for the genus *Plasmodium* was 42.9 years and women accounted for most cases with 82.4% (*n* = 14). Among the eight people who reported having malaria in the previous 12 months, two tested positive for the parasite by PET-PCR. Statistical analysis showed an association between previous cases in the house within the last 30 days and current malaria positivity (Fisher exact test *p* < 0.05, 95CI: 0.94–11.43, OR 3.3). No other epidemiological variables were associated with malaria positivity.

All 17 samples that returned at least one positive result for malaria (genus) were subjected to PET-PCR specific for either *P. falciparum* or *P. vivax*. Out of the total number of samples, nine (6.5% of the total and 52.9% of the total positive samples) revealed amplification for *P. falciparum*. Out of the positive results, eight had a Ct value below 40, while one amplification had a Ct value of 40.3. The mean Ct value for the amplifications of *P. falciparum* was 38.3 [37.3–40.3], with a standard deviation of 0.92. Additionally, seven samples yielded a positive result at least once for *P. vivax* (5% of the total, and 41.2% of the total of positive samples). The mean Ct value for the amplifications of *P. vivax* was 39.5 [36.6–40.8], with a standard deviation of 1.7. One of the samples exhibited a mixed infection by both parasite species. However, two samples that were found to be positive for the genus *Plasmodium* (11.8% of all positive samples) did not show any amplification for any of the two species (Figure 2). The nine samples that tested positive for *P. falciparum* were found to belong to the K1 allelic subfamily of the *Pfmsp1* gene.

## 4. Discussion

The study employed RDTs and ultrasensitive molecular approaches to assess the presence of asymptomatic malaria carriers in the Honduran Moskitia, a region with high malaria transmission rates (20 to 80 cases/1000 population). The PET-PCR analysis revealed that 8.7% of the subjects tested positive with a Ct value below 42, while 12% tested positive with a Ct value below 44. As expected, the PET-PCR method identified individuals with malaria parasitemia, while the RDTs were unable to detect it. Individuals were categorized as asymptomatic due to the absence of symptoms of malaria, despite having evidence of parasitemia. The parasitemias ranged between 100 parasites per μL and less than 0.25 p/μL. RDTs identified none of the cases, indicating that the method is less suited for this purpose. These results align with previous research that employed an active case detection strategy among people without symptoms, revealing positive cases despite a negative result from RDTs [10,12,21,22].

Asymptomatic infections behave like the submerged base of icebergs and are a hidden threat that hinders efforts aimed at eliminating malaria from the country and the Central American region. Infected individuals without symptoms are a source of persistent infection and contribute to the continued transmission of malaria [23,24]. Infections with very low parasite density are frequent in highly endemic settings in which the population has herd immunity and premunition. However, asymptomatic infections have also been demonstrated in regions of low transmission [25,26]. A reason for the absence of symptoms in some people infected with malaria could be attributed to the lower virulence of specific parasite strains, which allows them to avoid competition against more aggressive forms. According to the hypothesis proposed by Björkman, these less virulent strains could persist at low levels without being detected or treated for extended periods and are therefore subject to evolutionary selection [27].

Asymptomatic infections outnumber symptomatic ones in various malaria transmission settings [13]. Thus, the effectiveness of elimination strategies rests on the sensitivity of the diagnostic techniques, the results of which are then used to appropriately treat the carriers. The sensitivities of various malaria diagnostic methods have been compared in different epidemiological settings [28]. LM and RDTs have shown limited accuracy in detecting infections with low parasitemia, particularly in asymptomatic people, when compared to PCR [8,9,11,24,29,30,31,32,33,34,35].

Among the many variants of PCR-based methods, those that use PET-PCR have proven to be highly sensitive in detecting low parasitemias [6,7,15]. PET-PCR has been used mostly in the African context to evaluate the presence of asymptomatic carriers [8,9,10,11,12]. These authors compared PET-PCR with conventional diagnostic methods such as LM and RDT, agreeing that PET-PCR is much more sensitive than LM.

In the Americas, Lucchi et al. applied PET-PCR for the first time to detect falciparum malaria cases in a nationwide community survey in Haiti, with samples from almost 3000 individuals, of which 12 (0.4%) tested positive for *P. falciparum*. In that survey, the authors reported that 316 samples had a Ct value >40 but were considered negative as they did not amplify using two reference methods [7]. Valdivia et al. conducted a study in 217 participants in Peru in 2021 to assess the accuracy of a diagnostic instrument that relies on the detection of hemozoin and compared it against LM and two RDTs, using nPCR as a reference; PET-PCR was used to set cutoff points for the device and determined an optimal parasitemia limit for the device of 323 parasites/μL [36]. More recently, a study was carried out among 309 participants with malaria symptoms in Puerto Lempira, in the Honduran Moskitia, whose samples were analyzed by LM, nPCR, and PET-PCR. Malaria prevalence was 19.1% by LM and 31.1% by PET-PCR. These findings demonstrated that LM could identify 59 positive cases (19%), with a sensitivity of less than 60%, whereas molecular techniques (nPCR and PET-PCR) exhibited significantly higher sensitivity [6]. This study supports the findings of Matamoros et al., indicating that the diagnostic methods LM and RDT, commonly employed in Honduras for malaria detection, are inadequate in identifying cases of malaria with low parasitemias, particularly among asymptomatic carriers.

A significant finding from this survey is that the percentage of individuals with a positive PET-PCR result who reported living with someone diagnosed with malaria in the last 30 days is considerably higher (41.2%) compared to those with a negative result (10.7%). This finding corroborates the notion that a household in which one individual has been diagnosed with malaria is consequently more likely to have a cluster of infected individuals [37,38].

To evaluate the importance of asymptomatic individuals in the transmission of malaria, there are strategies such as cross-sectional active case detection (ACD) surveys [39,40,41]. These sampling strategies, when coupled with PCR-based tests, can be used to estimate the true burden of hidden infections [42]. These interventions could be regularly implemented in Honduras and other areas that aim to achieve malaria elimination by 2030. A good example of this strategy was applied by Contreras-Mancilla et al. in the Peruvian Amazon in 2018 [43]. The authors conducted four consecutive ACD interventions at 10-day intervals in four communities. The inhabitants were visited in their homes and an immediate diagnosis was made through LM. A quantitative PCR (qPCR) was used for subsequent molecular analysis, revealing that more than 54% of the participants tested positive for malaria, with most of them being asymptomatic.

Evidence suggests that people with asymptomatic malaria play a significant role in maintaining low levels of transmission in the community [13,24,38]. Consequently, novel strategies are necessary to successfully attain the objective of elimination [13,22]. The use of mass testing, treatment, and tracking (MTTT) seems to have contributed to the reduction of asymptomatic carriers among a population in Ghana [44]. Similarly, the mass screening and treatment (MSAT) strategy concentrates on providing antimalarial medications specifically to the population that is infected [45]. Alternatively, the approach called mass drug administration (MDA) for malaria, defined by the WHO as “the administration of a full therapeutic course of an antimalarial medicine at approximately the same time, and often at repeated intervals, to all age groups of a population in a defined geographical area. Antimalarial medicines are administered without prior malaria testing, and therefore, regardless of the malaria infection status of individuals. Consequently, any existing infections are treated and new infections are prevented for the duration of the drug’s prophylactic period” [3] which has demonstrated efficacy in the past for controlling and eliminating malaria and should be considered a component of a holistic approach for malaria elimination in particular contexts [46].

Some authors propose that MDA has the potential to reduce transmission for a limited time [47] and that it should not be used alone but together with other conventional strategies. At the same time, it must be implemented repetitively if it is to have a sustained effect [48]. According to the WHO, in situations where the prevalence of *P. falciparum* is below 10%, or the incidence is less than 250 cases per 1000 people annually, MDA may effectively decrease both the burden of the disease and the intensity of transmission [3].

The responsibility for designing and implementing new control and elimination strategies to target asymptomatic malaria reservoirs in Honduras will ultimately rest with the country’s decision-makers. Nevertheless, continuing to rely on traditional strategies like passive and reactive case detection after outbreaks emerge could hinder efforts to achieve the goal of malaria elimination by 2030.

## 5. Limitations

The main limitation in this study is the small sample size of individuals recruited, which means that the proportion of asymptomatic malaria carriers identified using this method does not accurately reflect the statistical prevalence of asymptomatic cases in Gracias a Dios, Honduras.

## 6. Conclusions

Despite the low number of participants recruited, this study provides insight into the prevalence of asymptomatic malaria infections at a high-transmission hotspot in the Americas. The low sensitivity of RDTs highlights the need to integrate molecular methods such as PET-PCR into national surveys carried out in the coming years. Finally, the presence of such a high number of asymptomatic reservoirs in the Moskitia region poses a hidden threat to malaria elimination efforts. These results highlight the importance of modifying the elimination strategies currently in force in the national standard since they are not capable of detecting or treating asymptomatic infected individuals promptly, which serve as reservoirs of the disease.

## Figures and Tables

**Figure 1 pathogens-13-00541-f001:**
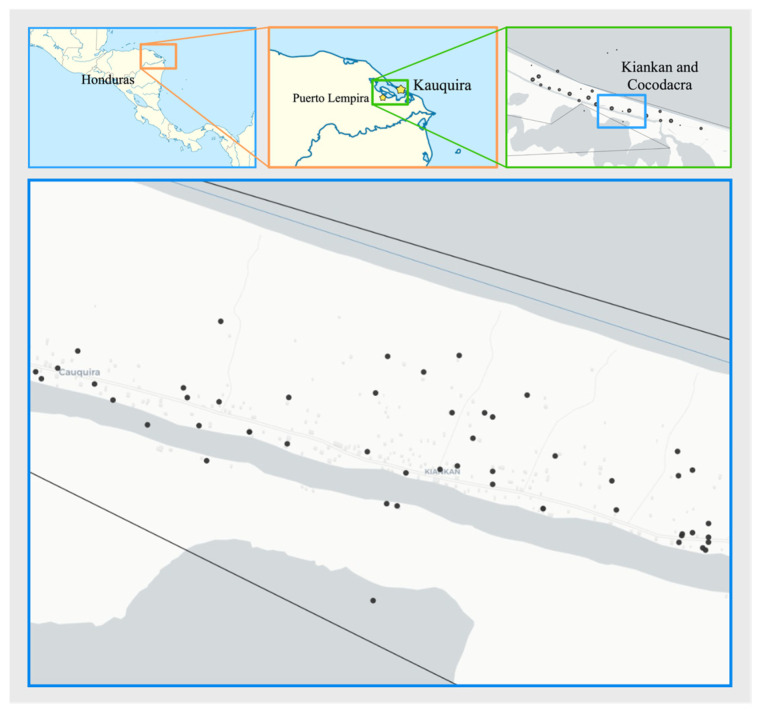
Map showing the neighbourhoods of Kiankan and Cocodacra, in the village of Kaukira, municipality of Puerto Lempira, where the active search for malaria cases was carried out. Black dots indicate households where health authorities reported index cases during 2023. Map created using open data from the GADM database of global administrative areas, version 3.6. www.gadm.org.

**Figure 2 pathogens-13-00541-f002:**
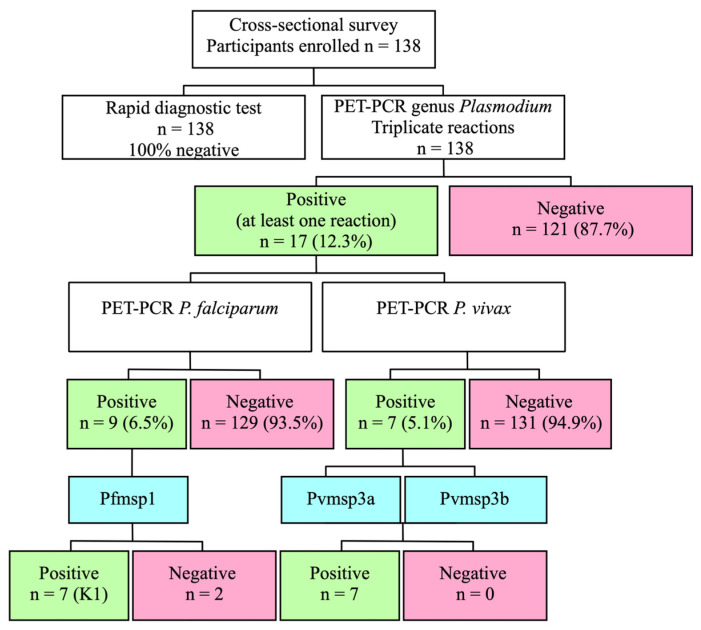
Workflow illustrating the findings of malaria testing techniques in a population without symptoms.

**Table 1 pathogens-13-00541-t001:** List of primers used for amplification reactions and nucleotide sequences.

Method and Target	Primer Name	Sequence (5′-3′)	References
PET-PCR for genus *Plasmodium*	Genus forward	GGC CTA ACA TGG CTA TGA CG	[6,15]
	Genus reverse	6FAM-agg cgc ata gcg cct gg CTG CCT TCC TTA GAT GTG GTA GCT	
PET-PCR for *P. falciparum*	Falciparum forward	ACC CCT CGC CTG GTG TTT TT	[15]
	Falciparum reverse	HEX-agg cgc ata gcg cct gg TCG GGC CCC AAA AAT AGG AA	
PET-PCR for *P. vivax*	Vivax forward	ACT GAC ACT GAT GAT TTA GAA CCC ATT T	[16]
	Vivax reverse	HEX-agg cgc ata gcg cct ggT GGA GAG ATC TTT CCA TCC TAA ACC T	
Nested PCR for *P. vivax msp3α* (1st round)	Pvmsp-3a P1	CAG CAG ACA CCA TTT AAG G	[17]
	Pvmsp-3a P2	CCG TTT GTT GAT TAG TTG C	
Nested PCR for *P. vivax msp3α* (2nd round)	Pvmsp-3a N1	GAC CAG TGT GAT ACC ATT AAC C	
	Pvmsp-3a N2	ATA CTG GTT CTT CGT CTT CAG G	
Nested PCR for *P. vivax msp3ß* (1st round)	Pvmsp-3b P1	GTA TTC TTC GCA ACA CTC	[17]
	Pvmsp-3b P2	CTT CTG ATG TTA TTT CCA G	
Nested PCR for *P. vivax msp3ß* (2nd round)	Pvmsp-3b N1	CGA GGG GCG AAA TTG TAA ACC	
	Pvmsp-3b N2	GCT GCT TCT TTT GCA AAG G	
Nested PCR for *P. falciparum msp1* (1st round)	MIOF	CTA GAA GCT TTA GAA GAT GCA GTA TTG	[18,19]
	MIOR	CTT AAA TAG ATT CTA ATT CAA GTG GAT CA	
Nested PCR for *P. falciparum msp1* (2nd round)	K1F	AAA TGA AGA AGA AAT TAC TAC AAA AGG TGC	
	K1R	GCT TGC ATC AGC TGG AGG GCT TGC ACC AG	
	MAD20F	AAA TGA AGG AAC AAG TGG AAC AGC TGT TAC	
	MAD20R	ATC TGA AGG ATT TGT ACG TCT TGA ATT ACC	
	RO33F	TAA AGG ATG GAG CAA ATA CTC AAG TTG TTG	
	RO33R	CAA GTA ATT TTG AAC TCT ATG TTT TAA ATC AGC GTA	

## Data Availability

All data generated or analyzed during this study are included in this published article.

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
