# Peer review of "Asymptomatic Malaria Reservoirs in Honduras: A Challenge for Elimination"

_pathogens, 2024, doi:10.3390/pathogens13070541_

Round 1
Reviewer 1 Report
Comments and Suggestions for Authors
Author Response
Dear Editors:
Regarding the manuscript titled "Asymptomatic malaria reservoirs in Honduras: a challenge for elimination", we would like to thank the editor and the reviewers for their valuable contributions.
We have read the comments and have provided a point-by-point response to each of them below. As a result, we have made the suggested changes to the document, which are highlighted in yellow for easy identification.
Sincerely,
Gustavo A. Fontecha S., PhD.
Point-by-point response
Reviewer 1
Review comments
This is an interesting study describing frequency of malaria in a region in Honduras using PET-PCR as diagnosis strategy, whose sensitivity is significantly higher. The article present two main results, the presence of P. falciparum and P. vivax in the country and the low sensitivity of RDT for low parasitaemias. However, the article ha some limitations related with its design (low sample size and it is not clear the study design) and with the discussion, more focus on comparison of techniques and drug administration strategies than in the proper discussion of the results.
Background
- Please include a reference in line 57, line 61 and line 62. It is essential to know what is the origin of that data.
Answer. I appreciate your observation. The Honduran Ministry of Health publishes weekly malaria epidemiology bulletins, which provide the data required in these calculations. The website where all of this information is available has been included.
- Line 73. It is true that all that groups could act as “hidden reservoirs” but it is important to highlight that the key point is the gametocytaemia to assess their contribution to transmission.
Answer. Thanks. We have emphasized the importance of gametocytemia by modifying the paragraph as follows: “These overlooked individuals serve as hidden reservoirs, as they may have gametocytes circulating in the peripheral blood, thus maintaining the transmission of the disease [5].”
- Line 84. I’m not sure why it is described the area a “different highly endemic region” if the previous paragraph is about Gracias a Dios department
Answer. The reviewer is correct. Thank you. We have modified the sentence as follows: “Considering the country’s entry into the elimination stage, this study aimed to determine the prevalence of asymptomatic malaria infections in La Moskitia region in Honduras, through an active search strategy”.
Methods
- Figure 1. I suggest to use more contrast in the map because it is complicated to interpretate
Answer. A new version of Figure 1 has been included in the manuscript. Thank you.
- Line 124. It could be useful to have more information about the process to choose the households included, as depends on their characteristics it could influence the results.
Answer. Unfortunately, we cannot say much more than what has been expressed. The researchers walked through the village, in which all the houses have similar conditions in appearance and socioeconomic level, and randomly approached the residents to talk about the project. There were no exclusion or inclusion criteria to select the houses. Those houses in which the inhabitants were willing to participate were included.
- Line 168. Please add the control used for P. vivax, I’m did not find it.
Answer. The same standard curve was used to calculate parasitemia for both parasite species.
Results
- Line 195. It is not clear how you decided the number of households and participants enrolled.
Answer. The number of participants was conditioned by the number of rapid diagnostic tests we had available (n= 150).
- Line 199. What “previous malaria case in the last 12 months”? The participant have had malaria in the last 12 months?
Answer. This is right. In the Moskitia region, with high endemicity, it is common for people to develop malaria once or twice a year.
- Line 207. As the authors ackwnoledge the parasitaemias were extremely low. In cases with Ct values over 42, maybe it should be confirmed the positive with duplicates.
Answer. Indeed, as indicated in section 2.4., all samples were analyzed at least in duplicate, and in some cases, they were analyzed 3 or 4 times. However, in some cases, due to low parasitemias, only one of the replicates returned a curve that could be interpreted as a positive result.
- Line 225. Did you perform any other test to know the specie of that samples?
Answer. To determine the species of Plasmodium causing the infection, PET-PCR was performed, which like any molecular diagnostic analysis is highly specific. So, it is not necessary to confirm with another test. In any case, the Pvmsp3 and Pfmsp1 markers were able to amplify 14 of the samples and confirm the parasite species.
Figure 2. The begging of the figure it is not clear, as it suggests that there were two different groups of 138 samples
Answer. It is possible that the reviewer misinterpreted the algorithm in Figure 2. What the figure indicates is that the 138 enrolled participants underwent both RDT and PET-PCR.
Discussion
- Line 241. It is completely true that the RDT were less sensible but I think that it is worthy to mention that they were also really useful for other conditions. Moreover, have you treat the positive samples? Or you could not find them again?
Answer. We absolutely agree with the reviewer. Rapid tests are an invaluable tool in the diagnosis and control of malaria, however one of the purposes of this work was to demonstrate that they do not fulfill their purpose in the context of asymptomatic or subclinical infections. On the other hand, none of the participants had a positive result in situ with RDT, and consequently they were not treated. The PET-PCR results were obtained months later and were communicated to the health authorities of the Gracias a Dios region for follow-up.
- Line 282. LM diagnosis was not described in the article, so it is not correct to include it in that sentence. Moreover, it could be recommended more analysis about diagnosis accuracy during results section, according to the importance given during discussion section.
Answer. The phrase in question summarizes the main findings of the publication by Matamoros et al, which is also from our research group. In that study, the sensitivity of LM versus nPCR and PET-PCR was compared. Although it is true that LM was not performed in this study, it is worth highlighting that the conventional techniques used in Honduras to diagnose malaria (LM and RDT) are not sufficient to detect asymptomatic or subclinical cases.
- Line 302 – 316. It could be useful to include what of these techniques is more recommended for Honduras and why.
Answer. In the conclusions it is established that: “The low sensitivity of RDTs highlights the need to integrate molecular methods such as PET-PCR in national surveys carried out in the coming years”
- It could be interesting to include a brief reflection on differences between P. falciparum and P. vivax.
Answer. We agree, however the scope of the work is already very broad, and including this reflection would not only lengthen the discussion, but would distance itself from the main objective of the manuscript.
Limitation
- Gametocytaemia is highly recommended.
Answer. We regret not understanding the meaning of the recommendation. We are open to considering any changes to the document if the suggestion is better explained to us.

Reviewer 2 Report
Comments and Suggestions for Authors
The authors want to analyze the presence of asymptomatic reservoirs in a location in Honduras, using a more sensitive method than RDT.
MAJOR COMMENT
As the authors themselves state in the paper, the main limitation of this study is the small size of the sample of individuals recruited, furthermore, there are also really many CTs to consider a positive sample. It would be useful to add a control population with an API rate <1, to understand how reliable these values obtained with PET-PCR .
MINOR COMMENTS
Table 1 is unclear, the primers in the various steps are not well identified, it would be better to divide them based on the type of analysis carried out
Figure 2 is unclear, and not only positive samples (n=17) are retested as indicated by the positive samples branch.
The authors of reference 4 are almost the same as this article, it would be correct to indicate this in the text.
Author Response
Reviewer 2
Comments and Suggestions for Authors
The authors want to analyze the presence of asymptomatic reservoirs in a location in Honduras, using a more sensitive method than RDT.
MAJOR COMMENT
- As the authors themselves state in the paper, the main limitation of this study is the small size of the sample of individuals recruited, furthermore, there are also really many CTs to consider a positive sample. It would be useful to add a control population with an API rate <1, to understand how reliable these values obtained with PET-PCR .
Answer. Without a doubt, it would be very interesting to analyze populations with low malaria endemicities. In fact, we have done it in the past with the school population (doi:10.1371/journal.pntd.0003248) and among pregnant women and their newborns
(Rev Panam Enf Inf 2018; 1(2):68-72). In both cases we did not find asymptomatic infections in regions with API rate <1. On the other hand, this study focused on the region of the country that contributes to more than 98% of the cases, so it is this region where it is pertinent to carry out this type of study.
MINOR COMMENTS
- Table 1 is unclear, the primers in the various steps are not well identified, it would be better to divide them based on the type of analysis carried out.
Answer. We have modified the table to give it greater clarity.
- Figure 2 is unclear, and not only positive samples (n=17) are retested as indicated by the positive samples branch.
Answer. Thank you so much. We have modified figure 2 in order to give it greater clarity and precision.
- The authors of reference 4 are almost the same as this article, it would be correct to indicate this in the text.
Answer. The following sentence has been modified: “A recent study was conducted by our research group, at Puerto Lempira, located in the Gracias a Dios department, to evaluate the sensitivity of LM compared to two molecular techniques”

Round 2
Reviewer 2 Report
Comments and Suggestions for Authors
Good job!